# Involvement of the Superior Colliculus in SIDS Pathogenesis

**DOI:** 10.3390/biomedicines11061689

**Published:** 2023-06-11

**Authors:** Anna M. Lavezzi, Riffat Mehboob, Francesco Piscioli, Teresa Pusiol

**Affiliations:** 1“Lino Rossi” Research Center for the Study and Prevention of Unexpected Perinatal Death and SIDS, Department of Biomedical, Surgical and Dental Sciences, University of Milan, 20122 Milan, Italy; 2Lahore Medical Research Center and LMRC Laboratories, LLP, Lahore 54000, Pakistan; mehboob.riffat@gmail.com; 3Provincial Health Care Services, Institute of Pathology, Santa Maria del Carmine Hospital, 38068 Rovereto, Italy

**Keywords:** brainstem, neuropathology, SIDS, superior colliculus, prone sleeping position, maternal smoking

## Abstract

The aim of this study was to investigate the involvement of the mesencephalic superior colliculus (SC) in the pathogenetic mechanism of SIDS, a syndrome frequently ascribed to arousal failure from sleep. We analyzed the brains of 44 infants who died suddenly within the first 7 months of life, among which were 26 infants with SIDS and 18 controls. In-depth neuropathological investigations of serial sections of the midbrain showed the SC layered cytoarchitectural organization already well known in animals, as made up of seven distinct layers, but so far never highlighted in humans, albeit with some differences. In 69% of SIDS cases but never in the controls, we observed alterations of the laminar arrangement of the SC deep layers (precisely, an increased number of polygonal cells invading the superficial layers and an increased presence of intensely stained myelinated fibers). Since it has been demonstrated in experimental studies that the deep layers of the SC exert motor control including that of the head, their developmental disorder could lead to the failure of newborns who are in a prone position to resume regular breathing by moving their heads in the sleep-arousal phase. The SC anomalies highlighted here represent a new step in understanding the pathogenetic process that leads to SIDS.

## 1. Introduction

SIDS (sudden infant death syndrome) is defined as the sudden death of an infant under 1-year of age that remains unexplained after a thorough death scene investigation, including a complete autopsy and detailed clinical review [1]. This syndrome is still the main cause of death in infants under the age of 1, despite widespread campaigns to place infants on their backs for sleeping. Already in 1992, the American Academy of Pediatrics (AAP) developed guidelines emphasizing the risks of prone sleep [2]. Subsequently, the National Institute of Child Health and Human Development (NICHD), in partnership with other pediatric associations, launched the *Back to Sleep* public education campaign that aimed to educate parents and caregivers to place infants to sleep in a supine position to reduce the risk of SIDS [3,4,5]. For the European Union, the ECAS (European Concerted Action on SIDS), on the basis of a large case-control study conducted in 20 European regions, emphasized the high and significant correlation between SIDS and the prone position in sleep in 1997. They, thus, laid the scientific foundations for the dissemination of information and the promotion of preventive action, i.e., the supine placement of babies, throughout European territory [6]. However, it is important to note that even if a newborn is placed in the supine position, it can roll over into the prone position, as this is an infant’s most spontaneous sleeping orientation [7].

The prone sleeping position can reduce oxygenation (hypoxia) and increase carbon dioxide levels (hypercapnia). This is exacerbated in the case of newborns who are regularly exposed to maternal cigarette smoke [8]. If an infant is healthy, in these conditions, he/she generally develops protective mechanisms including awakening and moving his/her head into a safe position to avoid rebreathing his/her own exhaled breath [9,10]. Contrastingly, a vulnerable newborn with structural brain abnormalities may fail to auto-resuscitate during a hypoxic episode because he/she is unable to move his/her head. This failure to respond in the presence of a hypoxic challenge can lead to SIDS [11,12]. In fact, numerous studies have demonstrated the association between morphological and/or functional alterations of the brainstem centers that regulate breathing/hypoxemia responses and SIDS [13,14,15,16,17,18].

The aim of this study was to investigate whether, in addition to the brainstem respiratory centers, other motor structures might also be involved in SIDS. We specifically focused our attention on the superior colliculus (SC), a paired laminar structure in the rostral midbrain where visual, auditory and somatosensory information are integrated to initiate specific motor commands to move the head, as well as the eyes [19,20,21]. More precisely, the SC is involved in integrating environmental stimuli in order to coordinate neck and eye muscle movements [22,23,24]. In addition, the decision to focus the study on the SC was due to the fact that there is considerable evidence of a connection between the respiratory centers and this structure, as it is able to initiate appropriate respiratory defensive responses in case of threatening stimuli [25,26] and to play a role in avoidance reactions [27,28,29]. Specifically, we investigated the possible presence of developmental alterations of the SC in a large set of SIDS cases compared to controls. Encouraging preliminary results were obtained, leading us to hypothesize that SC alterations can contribute to the pathogenetic mechanism of SIDS.

## 2. Materials and Methods

### 2.1. Study Subjects

The study included 44 infants, 19 females and 25 males, who died suddenly within the first 7 months of life. Following a thorough autopsy examination carried out in accordance with the guidelines provided by Italian law n.31/2006 “Regulations for Diagnostic Post Mortem Investigation in Victims of SIDS and Unexpected Fetal Death”. This law states that all infants suspected of SIDS, who died suddenly and unexpectedly within the first year of age without any apparent cause, must undergo an in-depth anatomo-pathological examination, including a detailed study of the autonomic nervous system. 

The cases were divided into two groups: (1) SIDS and (2) controls.

#### 2.1.1. SIDS Group

This group was composed of 26 cases, 11 females and 15 males (mean age: 3.4 postnatal months), for whom the cause of death remained unexplained after a thorough autopsy examination in accordance with the aforementioned law.

#### 2.1.2. Control Group

This group was composed of 18 cases, 8 females and 10 males (mean age: 3.5 postnatal months) for whom the cause of death was determined after a thorough examination. The specific diagnoses were infections (lower respiratory infections, blood infections) (n = 8), congenital heart defects (n = 4), disorders related to short gestation and low birthweight (n = 5), and toxic encephalopathy (n = 1). In none of the control cases was the cause of death related to previous prolonged hypoxia. Given that these cases were generally similar with regards to gender, ethnicity and age at the time of death to the SIDS cases, they were regarded as “controls”.

### 2.2. Risk Factor Information

For each case, all the available data regarding the pregnancy, delivery and environmental situation in which the death occurred was collected. Information on both the “preventable” and “unpreventable” risk factors, that are known to be associated with SIDS [30], was obtained and classified during post-mortem family interviews. Particular attention was paid to the preventable causes that could have been avoided, such as prone sleep positioning and maternal smoking during pregnancy. Smoking habits were divided into two categories: smokers vs. non-smokers. Regarding the sleeping position, the infants were divided into two groups: “found dead in prone position” and “found dead in supine position”.

### 2.3. Neuropathological Examination

An in-depth histological analysis of the central nervous system (CNS) was performed for all cases with the main aim of detecting fine developmental alterations in the SC, which is typically considered the major node for mediating sensorimotor commands for head turns, in addition to the structures usually analyzed (i.e. the hypoglossus, the dorsal motor vagal, the tractus solitarius, the ambiguous, the inferior olivary, the pre-Bötzinger, the arcuate, the dorsal and ventral cochlear nuclei, the medial and inferior vestibular nuclei, and the obscurus and pallidus raphé nuclei in the medulla oblongata; the locus coeruleus, the retrotrapezoid/parafacial complex, the superior olivary complex, the superior and lateral vestibular nuclei, the Kölliker-Fuse, the median and magnus raphé nuclei in the pons; the inferior colliculus, the substantia nigra, and the dorsal and caudal linear raphé nuclei in the caudal mesencephalon).

After removal, the brains were fixed in a 10% formalin solution for at least two weeks, then weighed and macroscopically evaluated. The study was specifically focused on the midbrain, with the region of interest being the SC. After cutting the cerebral peduncles, the midbrain was isolated from the brainstem. It was then cut into a sequential series of samples, orthogonal to the brainstem axis to allow sectioning in the axial plane. After rinsing these in running water to remove the formalin, they were dehydrated in ascending grades of alcohol, cleared with xylol, embedded in paraffin and sectioned using a microtome into 4 μm thick, serial sections. Sample sections were, subsequently, mounted on slides and stained with “hematoxylin/eosin” for a first observation under the microscope. This guided the selection of the most significant sections to be treated with the “Klüver–Barrera” staining method. The original method, set forth by Klüver and Barrera in 1953 [31], is in fact the most suitable for staining the human brainstem. The first step is staining with Luxol fast blue MBS (Color Index Solvent Blue 38, Sigma-Aldrich, code S3382), which reveals the myelinated fibers. The counterstain was carried out using cresyl violet to obtain a clear image of the nuclear chromatin in the neurons and glia. Further technical details can be found in our previous articles [32,33,34].

The microscopic examination was focused, in particular, on the cytoarchitecture of the SC in histological sections obtained consecutively from the rostral midbrain sample (Figure 1).

All of the observations were acquired with a Nikon Eclipse E800 light microscope (Nikon Corporation, Tokyo, Japan) and the images of the regions of interest were taken using a Nikon Coolpix 8400 digital camera attached to the microscope.

### 2.4. Statistical Methods

All the acquired images were independently analyzed by two pathologists who were blinded to the cause of death. The evaluations obtained from each observer regarding the various parameters were reported separately in a case report table. When the mean values had been calculated, they were compared using a K Index (KI) in order to check inter-observer reproducibility. The Landis and Koch [35] method was then used to interpret the K values (0 to 0.2 = slight agreement; 0.21 to 0.40 = fair agreement; 0.41 to 0.60 = moderate agreement; 0.61 to 0.80 = strong or substantial agreement; 0.81 to 0.99 = 1.00 = very strong or almost perfect agreement; 1.0 = perfect agreement). A very satisfactory KI value (0.89) was obtained in this study. The association between the groups, the superior colliculus (SC) alterations and the main risk factors was determined using the chi-square test. The statistical analyses were carried out using the SPSS (statistical package for social science) statistical software (SigmaPlot^®^, version 13, Systat Software Inc., Chicago, IL, USA). The associations were considered statistically significant if the *p* value was <0.05.

### 2.5. Consent

The parents of all of the SIDS cases and controls provided written informed consent for the autopsy and related research. Institutional review board approval was not required for this study since it complied with the requirements of Italian law n.31/2006.

For every case, a complete clinical history, including identification of the risk factors for sudden unexpected death, must be collected [36].

## 3. Results

In line with the aim of this study, the histological examination of the CNS was strongly focused on the SC.

### 3.1. Morphological Examination of the SC

The histological examination of the SC in serial sections of the rostral midbrain, where the red nucleus is clearly visible (Figure 1), revealed a normal multilaminar organization in all 18 controls (100%) and in 8 of the SIDS cases (31%). The classification scheme proposed by May [21], based on that of Kanesaki and Sprague [37], for various animal species (such as gray squirrel, macaque, feline, guinea pig) was used for the nomenclature of the layers. In human newborns, six of the seven layers May highlighted were easily identifiable, albeit with some differences in their cytoarchitecture. Listed from the outside, these layers are: the “stratum zonale” (SZ), a thin layer with scattered, very small cells; the “stratum griseum superficialis” (SGS), which has numerous small cells; and the “stratum opticum” (SO), which has a mixture of many darkly stained cells and numerous fibers. Beneath these “superficial layers”, which primarily receive visual sensory information, there are “deep layers” containing motor output neurons. In order, these consist of: the “stratum griseum intermedium” (SGI), with many larger, frequently multipolar neurons; the “stratum griseum profundum” (SGP), a layer with small, darkly stained neurons mixed with fibers; and the “stratum album profundum” (SAP), which is a fibrous thinner layer that lies immediately above the periaqueductal gray. Among the deep layers, the layer defined by May as the “stratum album intermedium” (SAI), located between the SGI and SGP, was not always identifiable in this human material (Figure 2).

In the remaining 18 SIDS cases (69%), the laminar organization displayed distinctly different characteristics. Three of these stood out. First, the individual strata were not as readily identified. This loss of clear lamination was largely due to the other two changes. Second, a considerable number of polygonal cells of the type normally seen in the SGI invaded the SGS and SO of the superficial layers (Figure 3). Third, there was an increase in the degree of staining of the gray layers (SGS, SGI and SGP) by Luxol fast blue (Figure 4). Thus, the neuronal processes in these layers contained more myelinated fibers. In particular, there was a dense intertwining of dark fibers mainly in the SGI (Figure 4).

In 10 of the 18 SIDS cases (56%), these SC alterations were associated with abnormalities in various brainstem centers that control respiratory activity and the sleep-arousing phase (especially hypoplasia of the medullary pre-Bötzinger nucleus, the pontine Kölliker-Fuse region and the pars compacta of the substantia nigra). In the other 8 SIDS cases (44%), the SC defects represented the only altered finding. In the remaining 8 SIDS cases with a normal SC structure, none of the aforementioned brainstem alterations were detected.

### 3.2. Correlation between SC Abnormalities and Preventable Risk Factors

Overall, 20 of the 44 mothers of the newborns (45%) smoked more than 3 cigarettes/day prior to pregnancy. Fifteen women in the smoker group were mothers of a SIDS victim and five were mothers of control cases. Nineteen of the 26 SIDS cases were found dead in the prone position, 6 were found on their backs (supine position) and 1 was found in a different situation (during the day, while awake in the stroller). Only 9 of the 18 controls died while sleeping, usually in the supine position (only one on its side). The other nine cases died in other situations. As regards the association with maternal smoking and sleeping position, 50% of the SIDS cases (9/18) with SC pathological alterations and a smoking mother died in a prone position. The other nine with SC alterations, all of which died in a prone position, were born from non-smoking mothers (Table 1).

Table 1 shows the association between the SC alterations and main risk factors among the groups. Categorical data are expressed as the number of cases. A strongly significant association was observed between the SC cytoarchitecture, maternal smoking and sleep position among both groups (*p*-value < 0.05). Regarding the association between SC defects and the main risk factors, nine cases of SIDS found dead in a prone position during sleep and with a smoking mother showed SC alterations (increased number of polygonal cells and myelinated fibers). The other nine SIDS cases with SC changes found in a prone position were born from non-smoking mothers.

## 4. Discussion

Our current knowledge of the anatomy and function of the SC has been acquired from experimental studies conducted on a range of animals. Experts in the field agree that the SC appears histologically as a laminated structure. King [38] and others have made a distinction between the “dorsal superficial layers”, which are exclusively visual, and the “deeper layers” that contain neurons that are responsive to visual, auditory and/or somatosensory stimuli. Based on their connections and cell activity, the seven layers, have been divided both histologically and connectionally into two groups: the “superficial layers” (the stratum zonale/SZ, the stratum griseum superficialis/SGS and the stratum opticum/SO), which are purely sensory and receive retinal input, and the “deep layers” (the stratum griseum intermedium/SGI, the stratum album intermedium/SAI, the stratum griseum profundum/SGP and the stratum album profundum/SAP) that receive inputs from both sensory and motor structures. Functionally, superficial layer cells are visually receptive, and the deep layer cells are able to integrate multisensory inputs to drive specific motor commands. Thus, the SC not only responds to retinal inputs from both the eyes, but also helps to orient the head and eyes towards visual, auditory and somatosensory stimuli [21,22,23,24]. This capability is especially important in early infancy during sleep awakening. Moreover, several studies have shown the involvement of the SC in infants’ cognitive, attentional and emotional behaviors, as well as in autism [39,40,41,42,43].

With this study we have filled a critical gap in our current knowledge of the structure and function of the SC in humans. It was observed that the cytoarchitecture of the human SC in human infants is similar to that of various mammals [21]. We identified six of the seven layers observed in other primates by May and Porter [44] and Perkins et al. [45], albeit with some differences, such as difficulty in detecting the SAI, as well as species specific variations in thickness. Hence, we can assume that the SC is a conserved brain structure in mammals even with variations across species. Furthermore, it appears to be fully developed, histologically, in normal infants. Our study, whose aim was to evaluate the possible involvement of the SC in SIDS pathogenesis, led us to identify alterations in its layers in approximately 70% of cases, which is a fairly striking finding. In fact, it is important to note that SIDS frequently occurs when the infant is in the awakening phase, especially when it is in a prone position with its nose and mouth perpendicular to the underlying pillow or mattress. This can occur even if it is placed on its back to sleep [7]. This situation, known as “rebreathing”, can quickly lead to low levels of oxygen and increased carbon dioxide levels. If the brain areas responsible for regulating awakening from sleep are well developed, in particular those that coordinate the movements required to react to a situation of this kind, like the SC, newborns employ protective mechanisms, such as turning their necks and heads into positions that free their airways in order to improve breathing. However, alterations, particularly of the SC deep layers, may lead to the failure of the newborns to resume regular breathing by moving their heads. This inability to respond could lead to hypoxia and death.

The probability of this outcome increases if there are alterations in the brainstem centers that control breathing and/or if the newborn is regularly exposed to cigarette smoke. Our findings support the many studies that have previously demonstrated this linkage.

### 4.1. Smoking, a Universally Recognized Risk Factor for SIDS

It has long been known that exposure to cigarette smoke, particularly maternal smoking during and after pregnancy, is a major environmental risk factor for SIDS [46,47,48,49], as it can lead to severe hypoxia caused by inhaling carbon monoxide (CO), a combustion product from cigarette smoke and a harmful toxic gas. When inhaled, CO enters the lungs where it binds to the hemoglobin in the red blood cells 200 to 300 times more avidly than oxygen and forms “carboxyhemoglobin” [50,51,52]. This complex is unable to release oxygen into the tissues, leading to systemic hypoxia, which seriously affects the most important organs, especially the brain [53]. Furthermore, if associated with prone sleeping, this situation can reduce the newborn’s spontaneous arousability and result in the failure to auto-resuscitate.

### 4.2. Interpretation of the Histological Changes

The anomalies in the SC layers observed in the majority of SIDS victims (and not in the controls) essentially consisted in the presence of an increased number of polygonal cells above the SGI and by the intense Luxol staining of the neuronal processes, mostly in the SGS, SGI and SGP. Both of these alterations have two possible interpretations: (1) the abnormalities in their normal structure could be the cause of the newborn’s inability to turn his/her head upon awakening in order to breathe; or (2) they may represent an attempt to compensate for abnormalities in the respiratory activity by increasing the connectivity of the SC with other structures. In this regard, the presence of many larger multipolar neurons, more superficial than those normally found in the SGI, is noteworthy. In the SGI, these multipolar cells represent motor output neurons that project to the brainstem centers for head and eye movement [21]. Their presence outside of the SGI might suggest a poorly organized motor output layer, supporting the first interpretation, or their presence may be aimed at increasing the number of inputs to these cells, supporting interpretation 2.

Similarly, the greater evidence of myelinated fibers in multiple SC layers where they are not usually observed, especially in the SGI, highlighted by their intense staining with Luxol fast blue, is striking. The SGI is thought to be involved in triggering defensive movements, which certainly includes head movements in response to airway compromise. It seems likely that the presence of these myelinated fibers is due to a loss of proper organization in this layer. This would support interpretation 1. However, it is possible that it is due to other alterations of the neuropil to produce Luxol staining that is not myelin-related, and that are caused by reorganization aimed at compensating for changes in the downstream respiratory centers [54], supporting interpretation 2. Thus, determining the underlying causes of the collicular changes will require further investigation.

If the infant is sleeping in the prone (face down) position and their mouth is in contact with soft bedding, it must not only be able to awaken and turn its head, but also to resume a normal respiratory rhythm under the control of the brainstem respiratory network. With this in mind, Thach et al. [55] performed a series of experiments on healthy, sleeping infants to demonstrate that arousal begins with, in addition to eye opening and head turning, a sigh, i.e., an augmented breath consisting of a strong inhalation followed by a prolonged exhalation to restore a normal respiratory pattern. Noteworthy, is the article by Müller-Ribeiro et al. [25], which demonstrates the involvement of the SC in initiating increased respiratory activity, thanks to its connections with the brainstem respiratory centers. Previously, Keay and colleagues [26] reported in an experimental study on rats that cells within the SC, when activated by electrical stimulation and/or microinjections of glutamate, are able to influence respiration by increasing the rate and depth of breath. These studies further strengthen our hypothesis that SC abnormalities may be closely involved in the pathogenetic mechanism of SIDS.

Previously we have highlighted the presence of developmental alterations of the inferior colliculus (IC), a well-known relay station for auditory pathways located caudally to the SC, in SIDS, also attributing to this structure a role in respiratory control, especially during the sleep–wake cycle [56,57]. In all cases from this study, however, the IC was found to be normally developed.

Furthermore, the fact that the SC has numerous connections with the forebrain regions [21,44,45,58], supports its possible involvement in cognitive, attentional and emotional behaviors. Collicular forebrain pathways also support alerting behaviors that may underlie arousal mechanisms critical to avoiding SIDS. Finally, these circuits may underlie a further key role for this structure in the pathogenesis of various neurological disorders (such as autism spectrum disorders (ASD), attention deficit hyperactivity disorder (ADHD) and Rett syndrome) [59,60,61].

In short, we believe that the SC is not simply a brain structure that transforms sensory signals into motor commands, but also that any SC abnormality could have significant neurodevelopmental consequences that could contribute to the pathogenesis of SIDS. More specifically in this regard, it can be hypothesized that the SC alterations could block the normal process of arousal that reverses the inactivation of skeletal motor activity, which occurs with sleep, in particular REM sleep. It is our intention to conduct further research on this topic with the aim of identifying developmental alterations of the SC under different neuropathological conditions. In addition, we intend to investigate the role of genes that regulate neural development and the cytoarchitectural configuration of specific brain structures, in particular of the SC, such as Pax7, in SIDS and in the aforementioned neuronal disorders [62].

Finally, the eight cases (about 30% of the case study) in which no neuropathological alterations were observed deserve consideration. We can hypothesize that the cause of death in these cases in which a neuroanatomical elucidation cannot be provided is multifactorial. The leading theory to explain SIDS in similar cases is the “Triple Risk Model” initially described by Filiano and Kinney in 1994 [63] and which is defined as three specific factors coming together to cause death. The three factors are: a vulnerable infant, a critical developmental period (2–4 months) and exogenous stressors (like exposure to cigarette smoking during and after pregnancy, soft mattresses, bedding and objects in the bed, co-sleeping, etc.).

## 5. Conclusions

The alterations of the SC detected in this study in most cases of SIDS demonstrate a failure to properly organize the colliculus into myelinated and cell-rich layers and, therefore, a deep disorder of the normal developmental programs in this syndrome. These anomalies represent a new step in understanding the pathogenetic process that leads to SIDS, allowing us to envisage that alone or in association with other developmental defects of the brainstem, they can determine the sudden death of an infant in the first months of life.

When a plausible case of death due to SIDS occurs, we believe that it is essential to conduct an in-depth analysis of the central nervous system, particularly the brainstem where the main structures involved in controlling vital activities are located, in addition to a routine autopsy examination aimed at ruling out an actual cause of death. This comprehensive examination, if conducted globally by expert neuropathologists, will help to identify specific developmental alterations of one or more nerve centers underlying these deaths. By correlating these defects with the main exogenous risk factors, it will be possible to explain the pathogenetic mechanism of SIDS and design preventive strategies to decrease the incidence of these distressing events for both parents and clinicians.

## Figures and Tables

**Figure 1 biomedicines-11-01689-f001:**
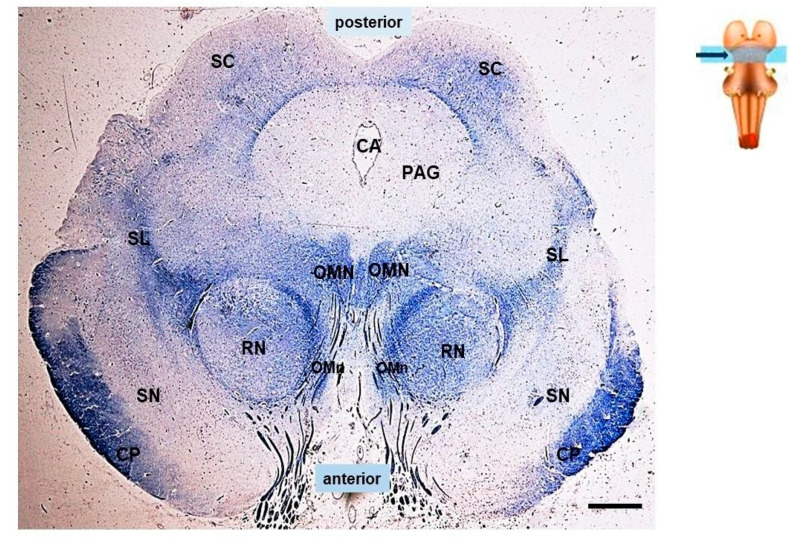
Transversal section of the rostral midbrain at the level of the red nucleus. Arrows indicate the location of the superior colliculi. CA: cerebral aqueduct; CP: cerebral peduncle; OMN: oculomotor nucleus; OCn: oculomotor nerve; PAG: periaqueductal gray; RN: red nucleus; SC: superior colliculus; SL: spinal lemniscus; SN: substantia nigra. Top right: brainstem diagram with indication of the location of the midbrain in the brainstem. Klüver–Barrera staining. Scale bar 1000 μm.

**Figure 2 biomedicines-11-01689-f002:**
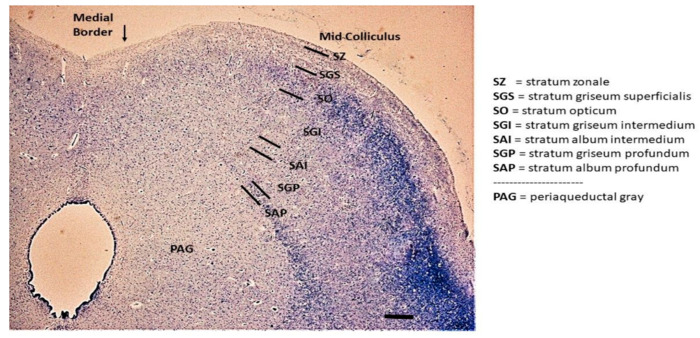
Cytoarchitectural laminar organization of the SC in the infant. The superficial layers consist of in order from the outside: the stratum zonale (SZ), the stratum griseum superficialis (SGS) and the stratum opticum (SO). The deep layers consist of: the stratum griseum intermedium (SGI), the stratum album intermedium (SAI), the stratum griseum profundum (SGP) and the stratum album profundum (SAP). PAG: periaqueductal gray. Klüver–Barrera staining. Scale bar 200 μm.

**Figure 3 biomedicines-11-01689-f003:**
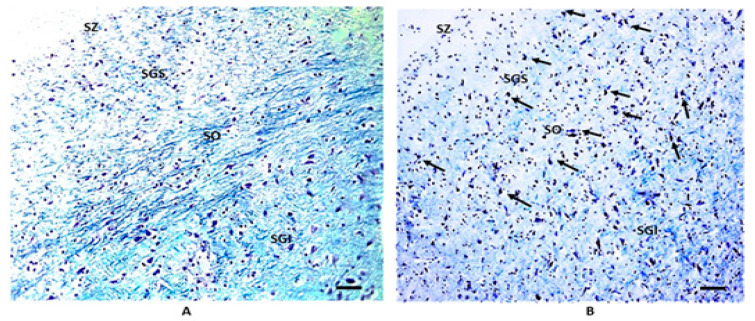
SIDS-related changes in cell lamination. (**A**) normal structure of the SC in a control case (male, 4 months); (**B**) a considerable number of polygonal cells invading the superior layers in a SIDS case (male, 4 months). Arrows indicate some of these neurons. Klüver–Barrera staining. Scale bar 100 μm.

**Figure 4 biomedicines-11-01689-f004:**
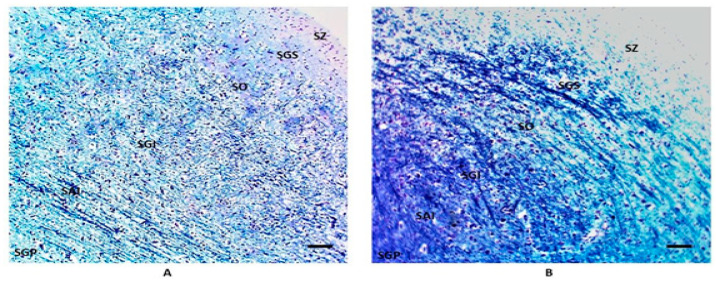
SIDS-related changes in myelin staining. (**A**) normal structure of the SC in a control case (female, 4 months); (**B**) a significant increase in the degree of myelin staining of the neuronal processes in the SGS, SGI and SGP in a SIDS case (male, 4 months). Klüver–Barrera staining. Scale bar 100 μm.

**Table 1 biomedicines-11-01689-t001:** Association between superior colliculus (SC) alterations and the main risk factors among the groups.

Variables	SIDS	Control	Total	*p*-Value
**SC cytoarchitecture**				
**Normal**	8	18	26	<0.05
**Alterations**	18	0	18	
**Maternal smoking**				
**Smoker**	15	3	18	0.006
**Non-smoker**	11	15	26	
**Sleep position**				
**Prone**	19	0	19	<0.05
**Supine**	6	8	14	
**On their side**	0	1	1	
**Other situation**	1	9	10	

## Data Availability

Not applicable.

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
