# Peer review of "Involvement of the Superior Colliculus in SIDS Pathogenesis"

_biomedicines, 2023, doi:10.3390/biomedicines11061689_

Round 1

Reviewer 1 Report

Sudden Infant Death Syndrome is an important issue for both science and medicine. In the manuscript entitled „Involvement of the superior colliculus in SIDS pathogenesis“ the authors attempted to analyse involvement of the mesencephalic superior colliculus in the pathogenetic mechanism of SIDS. The chosen methods of analysis are appropriate for this type of study. They are sufficiently described and explained. Neuropathological investigations of serial sections of the midbrain for the first time showed the  superior colliculus cytoarchitectural organization in humans. The Authors shoved alterations of the laminar arrangement of the superior colliculus deep layers in SIDS pacients. The obtained results are clearly presented. Discussion is broad and meaningful. In my opinion it is suitable for publication.

Author Response

Response to Reviewer 1 Comments

Sudden Infant Death Syndrome is an important issue for both science and medicine. In the manuscript entitled „Involvement of the superior colliculus in SIDS pathogenesis“ the authors attempted to analyse involvement of the mesencephalic superior colliculus in the pathogenetic mechanism of SIDS. The chosen methods of analysis are appropriate for this type of study. They are sufficiently described and explained. Neuropathological investigations of serial sections of the midbrain for the first time showed the  superior colliculus cytoarchitectural organization in humans. The Authors shoved alterations of the laminar arrangement of the superior colliculus deep layers in SIDS patients. The obtained results are clearly presented. Discussion is broad and meaningful. In my opinion it is suitable for publication.

Reply- Thank you very much for the positive evaluation.

Reviewer 2 Report

In the current study, Lavezzi et al. explored histological changes associated with sudden infant death syndrome (SIDS) in the mesencephalic superior colliculus (SC) in postmortem materials. The authors obtained highly interesting and novel results demonstrating that SIDS is accompanied by the invasion of polygonal cells into the stratum opticum and stratum griseum superficialis, as well as an increase in the density of myelinated fibers in the stratum griseum intermedium. Observed changes can underlie the mechanism of SIDS induction; however, this issue should be further carefully investigated. Given the exciting and novel findings described in the work, I think that this manuscript can be published in Int. J. Mol. Sci., however, only in the format of a Short Communication.

Major comments:

1. Authors draw conclusions based on the results of histological studies only, without proper statistical analysis. According to Section 2.4, the authors used ANOVA, however, I did not find any indication of p-values in the Results. Dear authors, please try to perform quantitative analysis of your histologic data.

2. The values depicted in Table 1 demonstrated only the characteristics of patients, but not the correlation between them. Please perform a correct correlation analysis of the data.

3. As mentioned above, the manuscript needs to be rewritten in accordance with the style of a Short Communication. The absence of important characteristics such as statistical analysis, double-check procedures, and evaluation of findings by other methods does not allow this work to be considered a full-length research article.

Minor comments:

1. Please add scale bars to Fig. 1 and 2.

2. Please correct the links to references according to the journal's rules.

Author Response

Response to Reviewer 2 Comments

In the current study, Lavezzi et al. explored histological changes associated with sudden infant death syndrome (SIDS) in the mesencephalic superior colliculus (SC) in postmortem materials. The authors obtained highly interesting and novel results demonstrating that SIDS is accompanied by the invasion of polygonal cells into the stratum opticum and stratum griseum superficialis, as well as an increase in the density of myelinated fibers in the stratum griseum intermedium. Observed changes can underlie the mechanism of SIDS induction; however, this issue should be further carefully investigated. Given the exciting and novel findings described in the work, I think that this manuscript can be published in Int. J. Mol. Sci., however, only in the format of a Short Communication.

Major comments:
  1. Authors draw conclusions based on the results of histological studies only, without proper statistical analysis. According to Section 2.4, the authors used ANOVA, however, I did not find any indication of p-values in the Results. Dear authors, please try to perform quantitative analysis of your histologic data.

Reply - An appropriate statistical analysis has been made with the indication of p-vaues.

  1. The values depicted in Table 1 demonstrated only the characteristics of patients, but not the correlation between them. Please perform a correct correlation analysis of the data.

Reply - This has been made.

  1. As mentioned above, the manuscript needs to be rewritten in accordance with the style of a Short Communication. The absence of important characteristics such as statistical analysis, double-check procedures, and evaluation of findings by other methods does not allow this work to be considered a full-length research article.

Reply -We performed an appropriate statistical analysis. As for the invitation to reduce our manuscript to a “Short communication”, we feel we have to disagree. First of all this contrasts with what is reported in the comments of Reviewers 3 and 4 who instead suggest clarifications and lengthening of the discussion by adding further considerations.

We also allow ourselves to highlight that this study was conducted under the supervision of Prof. Paul J. May of the University of Mississippi, one of the greatest experts of superior colliculus, albeit in experimental studies. He enthusiastically followed our research and the drafting of this manuscript since modifications of the SC in humans and in particular in infants who died of SIDS can be reported for the first time in the literature. For this reason we believe that the manuscript should not be reduced to a simple Short Communication.

Minor comments:
  1. Please add scale bars to Fig. 1 and 2.

Reply - Incomplete versions of figures 1 and 2 were submitted by mistake. They have been replaced.    

  1. Please correct the links to references according to the journal's rules.

Reply - This has been done.

Reviewer 3 Report

The article is focused on the important problem of searching for morphological brain correlates of Sudden Infant Death Syndrome (SIDS). The morphological changes detected in the superior colliculus (SC) in 69% of the brain samples of infants who died of SIDS are of undoubted scientific interest.

However, the characteristics of the groups with SIDS and the control group are not clearly defined.

Since the authors attribute the main pathogenesis of the detected morphological changes in the brain tissue area in infants with SIDS to prolonged tissue hypoxia, the choice of the control group needs justification. The control group should have been expected to include deaths not associated with previous prolonged hypoxia, which could have similarly affected the morphogenesis of the detected changes in the superior colliculus. The control group appeared to be very mixed in causes of death, which could be based on both acute conditions (infections suffered, toxic encephalopathy) and long-term hypoxic conditions (congenital malformations, body weight deficiency, blood infections?). Consequently, the identified causes of morphofunctional changes in the cell layers of the superior colliclus in infants with SIDS can no longer be explained solely by the causes of prolonged fetal hypoxia. Genetic defects may have played the leading role here.

 Table 1 should have a more precise name, since the correlation values as a statistical feature are not presented here. Separate data should be given here for both the SIDS group and the control group on the number of individuals who died in different position (prone, supine ets.), infants from smoking and non-smoking mothers, and SC cytoarchitecture. Also, the table should present statistically significant differences between the groups to prove and substantiate the role of the smoking factor and body position at time of death in both the SIDS group and the control group.

A limitation of the study in the form of an insufficiently well-matched control group for cause of death should probably be introduced in the Discussion section.

Author Response

Response to Reviewer 3 Comments

The article is focused on the important problem of searching for morphological brain correlates of Sudden Infant Death Syndrome (SIDS). The morphological changes detected in the superior colliculus (SC) in 69% of the brain samples of infants who died of SIDS are of undoubted scientific interest.
However, the characteristics of the groups with SIDS and the control group are not clearly defined.Since the authors attribute the main pathogenesis of the detected morphological changes in the brain tissue area in infants with SIDS to prolonged tissue hypoxia, the choice of the control group needs justification. The control group should have been expected to include deaths not associated with previous prolonged hypoxia, which could have similarly affected the morphogenesis of the detected changes in the superior colliculus. The control group appeared to be very mixed in causes of death, which could be based on both acute conditions (infections suffered, toxic encephalopathy) and long-term hypoxic conditions (congenital malformations, body weight deficiency, blood infections?). Consequently, the identified causes of morphofunctional changes in the cell layers of the superior colliclus in infants with SIDS can no longer be explained solely by the causes of prolonged fetal hypoxia. Genetic defects may have played the leading role here.

Reply- The authors thank the reviewer for this comment which allowed us to clarify. What we mainly wanted to highlight is the association between the development changes of the SC and the prone position. Since the deep layers of the SC control the movements of the head, when the newborn  is in a prone position he must turn its head to increase the respiratory rhythm, slowed down during his sleep.  But if these layers are altered, the newborn is not able to move the head and then he cannot avoid rebreathing his own exhaled breath being in a hypoxic state. 

The control cases we considered, irrespective of cause of death,  did not have morphological alterations of the SC and had not died in the prone position. Anyway, in none of the control cases the cause of death was related to previous prolonged hypoxia (this was added in the manuscript). If, in any event, a hypoxic condition occurred concurrently with the true cause of death, it did not affect the development of SC. We therefore believe that the case controls of the study can be adequate.

 Table 1 should have a more precise name, since the correlation values as a statistical feature are not presented here. Separate data should be given here for both the SIDS group and the control group on the number of individuals who died in different position (prone, supine ets.), infants from smoking and non-smoking mothers, and SC cytoarchitecture. Also, the table should present statistically significant differences between the groups to prove and substantiate the role of the smoking factor and body position at time of death in both the SIDS group and the control group.
A limitation of the study in the form of an insufficiently well-matched control group for cause of death should probably be introduced in the Discussion section.

Reply - More precise statistical evaluations have been introduced and, as previously indicated, we believe that the study's case controls are adequate.

Reviewer 4 Report

Authors present a retrospective study on 44 infants who died within first 7 months of life due to sudden infant death syndrome (SIDS) in 26 patients and 18 controls to investigate potential role of superior colliculus in the pathogenesis. In 69% of SIDS and none controls alterations of the laminar arrangement of the SC deep layers were observed; the hypothesis of the authors is that the the deep layers of the SC exert motor control including that of the head and that their developmental disorder could lead to failure of the newborns who are in a prone position to resume regular breathing by moving their heads in the sleep-arousal phase.

Authors continue their series on publications related to alterations of midbrain structures in context of SIDS; a bit peculiar is that there are no further publications from other research groups on this subject. What is even more peculiar is that the authors do not mention their previous work published in 2015: 

Lavezzi AM, Pusiol T, Matturri L. Cytoarchitectural and functional abnormalities of the inferior colliculus in sudden unexplained perinatal death. Medicine (Baltimore). 2015 Feb;94(6):e487. doi: 10.1097/MD.0000000000000487. PMID: 25674737; PMCID: PMC4602737. Lavezzi AM, Ottaviani G, Matturri L. Developmental alterations of the auditory brainstem centers--pathogenetic implications in Sudden Infant Death Syndrome. J Neurol Sci. 2015 Oct 15;357(1-2):257-63. doi: 10.1016/j.jns.2015.07.050. Epub 2015 Aug 1. PMID: 26254624.   Now, this required an explanation - is the current work continuation of the previous ones? Are these the same patients involved? Why is it now that superior colliculus is important for SIDS and not the inferior colliculus? Why did the authors investigated the relevance in only SC and not IC? I suggest to bring the current results into the context of research of the entire working group.  The conclusions seem to be too circumstantial and I suggest to make these in the form of educated guesses or proposals. I suggest to include clinical aspects and possible clinical applications of this study. 

Acceptable. 

Author Response

Response to Reviewer 4 Comments

Authors present a retrospective study on 44 infants who died within first 7 months of life due to sudden infant death syndrome (SIDS) in 26 patients and 18 controls to investigate potential role of superior colliculus in the pathogenesis. In 69% of SIDS and none controls alterations of the laminar arrangement of the SC deep layers were observed; the hypothesis of the authors is that the the deep layers of the SC exert motor control including that of the head and that their developmental disorder could lead to failure of the newborns who are in a prone position to resume regular breathing by moving their heads in the sleep-arousal phase.
Authors continue their series on publications related to alterations of midbrain structures in context of SIDS; a bit peculiar is that there are no further publications from other research groups on this subject. What is even more peculiar is that the authors do not mention their previous work published in 2015: 
Lavezzi AM, Pusiol T, Matturri L. Cytoarchitectural and functional abnormalities of the inferior colliculus in sudden unexplained perinatal death. Medicine (Baltimore). 2015 Feb;94(6):e487. doi: 10.1097/MD.0000000000000487. PMID: 25674737; PMCID: PMC4602737. Lavezzi AM, Ottaviani G, Matturri L. Developmental alterations of the auditory brainstem centers--pathogenetic implications in Sudden Infant Death Syndrome. J Neurol Sci. 2015 Oct 15;357(1-2):257-63. doi: 10.1016/j.jns.2015.07.050. Epub 2015 Aug 1. PMID: 26254624.   
Now, this required an explanation - is the current work continuation of the previous ones? Are these the same patients involved? Why is it now that superior colliculus is important for SIDS and not the inferior colliculus? Why did the authors investigated the relevance in only SC and not IC? I suggest to bring the current results into the context of research of the entire working group.  The conclusions seem to be too circumstantial and I suggest to make these in the form of educated guesses or proposals. I suggest to include clinical aspects and possible clinical applications of this study. 

Reply - The cases of SIDS considered here are different from those included in the previous studies cited above. As can be read in the Results, the SC alterations were accompanied in about half of the cases by development anomalies of other brainstem structures (precisely hypoplasia of the medullary pre-Bötzinger nucleus, the pontine Kölliker-Fuse and of pars compacta of the substantia nigra). Therefore, in no case were anomalies of the inferior colliculus (IC) detected. For this reason we did not cite our previous studies on IC. However, as rightly suggested, we have introduced this reminder in the Discussion (“Previously we have highlighted the presence of developmental alterations of the inferior colliculus (IC), a well known relay station for auditory pathways located caudally to the SC, in SIDS, attributing to this structure a role also in respiratory control, especially during the sleep-wake cycle [56,57].  In all cases of this study, however, the IC was found normally developed”).

We greatly appreciated the latest suggestions from the Reviewer (The conclusions seem to be too circumstantial and I suggest ……which led us to reveal the indications that we would like to give to all anatomopathologists and neuropathologists. We have therefore added these sentences at the end of Conclusions: “When a plausible case of death due to SIDS occurs, we believe that it is essential to conduct an in-depth analysis of the central nervous system, particularly the brainstem where the main structures involved in controlling vital activities are located, in addition to the routine autopsy examination aimed at ruling out an actual cause of death. This comprehensive examination, if conducted globally by expert neuropathologists, will help to identify specific developmental alterations of one or more nerve centers underlying these deaths.  By correlating these defects with the main exogenous risk factors, it will be possible to explain the pathogenetic mechanism of SIDS and design preventive strategies to decrease the incidence of these distressing events for both parents and clinicians.”

Round 2

Reviewer 2 Report

In the current study, Lavezzi et al. investigated the histological changes that are linked to sudden infant death syndrome (SIDS) in the mesencephalic superior colliculus (SC) using postmortem materials. The authors obtained highly interesting and novel results that demonstrate that SIDS is accompanied by the invasion of polygonal cells into the stratum opticum and stratum griseum superficialis, as well as an increase in the density of myelinated fibers in the stratum griseum intermedium. The authors addressed all the issues I identified in the first version of the manuscript submitted to Int. J. Mol. Sci. In my opinion, the supervision of Professor Paul J. May is not a strong enough reason to refrain from reformatting this article as a Short communication (full research articles require double-checking of results and the use of a wider range of experimental methods). However, considering the excellent quality of the Discussion and the requests made by other reviewers to expand some paragraphs in this section, I am forwarding my comment regarding the reformatting of the current manuscript to a Short Communication to the Editor. Little minor comment: Dear authors, please correct the p-values in Table 1. These values cannot be 0.000. In the case of extremely small p-values, you can use the following scientific notation: 3E-07.

Author Response

Dear Reviewer,

I personally thank you for your further comments on our article. I would like to better explain my reasons why our manuscript should be regarded as a research article and not as a short communication. 

This is not so much due to the fact that we used the advice of an expert to interpret both the normal and pathological findings we observed in the SC of newborns (not always evident as they appear in animals). I believe that this study not only has highlighted in a high percentage of infants who died from SIDS the presence of developmental alterations of a brain structure hitherto never considered as associated with this syndrome, but above all that it has provided for the first time in the literature a plausible explanation of why babies are found dead in their sleep in the prone position.

As described in depth in the Discussion, if a newborn with developmental SC abnormalities is sleeping in the prone position, given that this structure controls head/neck movements, he is unable in a hypoxic situation to move his head in order to breathe oxygen (inter alia, we are carrying out a retrospective study on numerous cases in the archive and the results seem to confirm our theory)

I hope you can take my thoughts into consideration.

Sincerely,

Anna Lavezzi

Reviewer 4 Report

Authors have sufficiently responded to reviewer remarks. 

Acceptable. 

Author Response

We thank the reviewer very much for the positive evaluation of the corrections we have made to the manuscript based on his comments.